# nnLandmark: A Self-Configuring Method for 3D Medical Landmark Detection

**Alexandra Ertl**[1,2,3] (iD)                                    ALEXANDRA.ERTL@DKFZ-HEIDELBERG.DE
**Stefan Denner**[1,4] (iD)                                        STEFAN.DENNER@DKFZ-HEIDELBERG.DE
**Robin Peretzke**[1,2,3] (iD)                                    ROBIN.PERETZKE@DKFZ-HEIDELBERG.DE
**Shuhan Xiao**[1,4] (iD)                                          SHUHAN.XIAO@DKFZ-HEIDELBERG.DE
**David Zimmerer**[1] (iD)                                         D.ZIMMERER@DKFZ-HEIDELBERG.DE
**Maximilian Fischer**[1,2] (iD)                           MAXIMILIAN.FISCHER@DKFZ-HEIDELBERG.DE
**Markus Bujotzek**[1,2] (iD)                               MARKUS.BUJOTZEK@DKFZ-HEIDELBERG.DE
**Xin Yang**[6] (iD)
**Peter Neher**[1,3,5] (iD)                                         P.NEHER@DKFZ-HEIDELBERG.DE
**Fabian Isensee**[*1,7] (iD)                                     F.ISENSEE@DKFZ-HEIDELBERG.DE
**Klaus H. Maier-Hein**[*1,2,3,4,5,7,8,9] (iD)           K.MAIER-HEIN@DKFZ-HEIDELBERG.DE

[1] *German Cancer Research Center (DKFZ) Heidelberg, Division of Medical Image Computing, Heidelberg, Germany*

[2] *Medical Faculty Heidelberg, Heidelberg University, Heidelberg, Germany*

[3] *Pattern Analysis and Learning Group, Department of Radiation Oncology, Heidelberg University Hospital, Heidelberg, Germany*

[4] *Faculty of Mathematics and Computer Science, Heidelberg University, Heidelberg, Germany*

[5] *German Cancer Consortium (DKTK), DKFZ, core center Heidelberg, Germany*

[6] *School of Biomedical Engineering, Shenzhen University Medical School, Shenzhen University, Shenzhen, Guangdong, China*

[7] *Helmholtz Imaging, DKFZ, Heidelberg, Germany*

[8] *National Center for Tumor Diseases (NCT), NCT Heidelberg, A Partnership Between DKFZ and The University Medical Center Heidelberg, Heidelberg, Germany*

[9] *HIDSS4Health, Heidelberg, Germany*

**Editors:** Under Review for MIDL 2026

## Abstract

Landmark detection is central to many medical applications, such as identifying critical structures for treatment planning or defining control points for biometric measurements. However, manual annotation is labor-intensive and requires expert anatomical knowledge. While deep learning shows promise in automating this task, fair evaluation and interpretation of methods in a broader context, are hindered by limited public benchmarking, inconsistent baseline implementations, and non-standardized experimentation. To overcome these pitfalls, we present nnLandmark, a self-configuring framework for 3D landmark detection that combines tailored heatmap generation, loss design, inference logic, and a robust set of hyperparameters for heatmap regression, while reusing components from nnU-Net's underlying self-configuration and training engine. nnLandmark achieves state-of-the-art

---

* Supervised equally

performance across three public and one private dataset, benchmarked against three recently published methods. Its out-of-the-box usability enables training strong landmark detection models on new datasets without expert knowledge or dataset-specific hyperparameter tuning. Beyond accuracy, nnLandmark provides both a strong, common baseline and a flexible, standardized environment for developing and evaluating new methodological contributions. It further streamlines evaluation across multiple datasets by offering data conversion utilities for current public benchmarks. Together, these properties position nnLandmark as a central tool for advancing 3D medical landmark detection through systematic, transparent benchmarking, enabling to genuinely measure methodological progress. The code will be available upon acceptance.

**Keywords:** 3D Medical Landmark Detection, Self-Configuration, Benchmarking.

## 1. Introduction

The task of medical landmark detection concerns the prediction of coordinates of predefined, anatomical keypoints. Accurate localization is critical for several medical imaging applications, including diagnosis, treatment planning, and navigation. In practice, these include the detection of anatomical reference points for image registration, control points for biometric measurements, small critical structures for surgical planning or fetal pose estimation in ultrasound (Taha et al., 2023; He et al., 2024; Chen et al., 2020; Gong et al., 2025c). Annotating such keypoints is highly dependent on detailed anatomical knowledge. For example, for fetal brain biometry, this requires reliably localizing the cerebellar landmarks, which is complicated by densely folded cortical structures and low tissue contrast (Gong et al., 2025c). Further, medical landmark annotation often involves between 10 and 50 landmarks, resulting in a time-consuming process, especially in 3D imaging data. Efforts in automating this task based on deep learning have already shown promising results (Schwendicke et al., 2021; Serafin et al., 2023; Singh et al., 2020). While earlier approaches aimed at directly predicting coordinate values, the current state-of-the-art formulation is heatmap regression. Thereby, each landmark is represented by a Gaussian-like blob in a dedicated output channel. During prediction, the coordinates are derived via channel-wise maximum (Payer et al., 2016; Pfister et al., 2015). The default base architecture for pixel-wise heatmap regression is the U-Net (Ronneberger et al., 2015; Çiçek et al., 2016). Many efforts have been made to optimize the architecture, aiming to effectively integrate global context or maintain high spatial resolution or super-resolution for sub-pixel accuracy of localization (Huang et al., 2025; Zhang et al., 2024). However, despite active methodological research in 3D medical landmark detection, progress is still affected by various pitfalls concerning benchmarking and usability of methods (Figure 1), identified from an extensive list of relevant, recent publications (Annex A).

**Pitfall 1: Insufficient public benchmarking.** The 3D landmark detection domain suffers from a lack of established and commonly used public benchmarks (Figure 1, left). Frequently, new developments only target single, often private datasets, leaving the generalizability of these methods to other tasks in question. This hinders a transparent interpretation of the results in a borader context and limits fair comparison to other methods (He et al., 2024; Schwendicke et al., 2021). While broad public benchmarking is already the standard in segmentation and has greatly propelled the field forward (Isensee et al., 2021, 2024), in the landmark detection domain, universal insights, generalizable solutions and the

broader impact of new developments are often left unexplored by focusing on single and private datasets.

**Pitfall 2: Inconsistent baseline implementations** Many publications compare their methods to a 3D U-Net baseline (Ronneberger et al., 2015; Çiçek et al., 2016). However, variations in hyperparameters, implementations, and training setups can substantially change performance, even when using the same underlying architecture. As depicted in Figure 1, center column, this is also evident for landmark detection based on reported U-Net results on the Mandibular Molar Landmark (MML) dataset, with mean radial errors (MRE) ranging from 1.9 mm to 2.7 mm (Huang et al., 2025; He et al., 2024; Zhang et al., 2024). In absence of a strong, commonly adopted baseline, the field lacks essential context for interpreting the results of new methods and assessing progress across datasets.

**Pitfall 3: Limited out-of-the-box usability.** The lack of comprehensive benchmarking and the use of non-standardized, custom code bases have led to dataset-specific implementations. Many methods are still published without code or clear instructions on how to adapt them to new datasets, for example when dealing with different modalities or image geometries (Figure 1, right). Applying such methods to new datasets can therefore require substantial expert knowledge in model development and resource-intensive hyperparameter tuning, which complicates broader application and increases the risk of reimplementation errors. The reliance on custom code further introduces potential confounding factors, obscuring the true performance of baseline architectures and leading to unclear conclusions about new developments. A standardized environment that works out-of-the-box across datasets is therefore crucial to enable transparent, systematic evaluation of new methods and quantify true methodological progress.

To counteract these pitfalls, we make the following contributions:

- We present a comprehensive benchmark study for 3D medical landmark localization, evaluating three recent state-of-the-art methods across four diverse datasets that span different imaging modalities and anatomical regions.

- We introduce **nnLandmark**, a fully self-configuring framework for 3D heatmap-based landmark detection that builds on the nnU-Net infrastructure to automatically derive dataset-specific preprocessing and training hyperparameters, enabling robust out-of-the-box generalization to new datasets without manual intervention.

- We show that nnLandmark consistently achieves state-of-the-art performance across all four benchmark datasets, surpassing existing methods and establishing a strong, reproducible baseline for future developments in 3D landmark detection.

- We demonstrate that nnLandmark serves as a flexible, standardized environment for method development by integrating the H3DE architecture into the framework, yielding clear performance gains over the official implementation and highlighting the value of leveraging a proven experimental infrastructure for systematic ablations and fair evaluation of new methodological contributions.

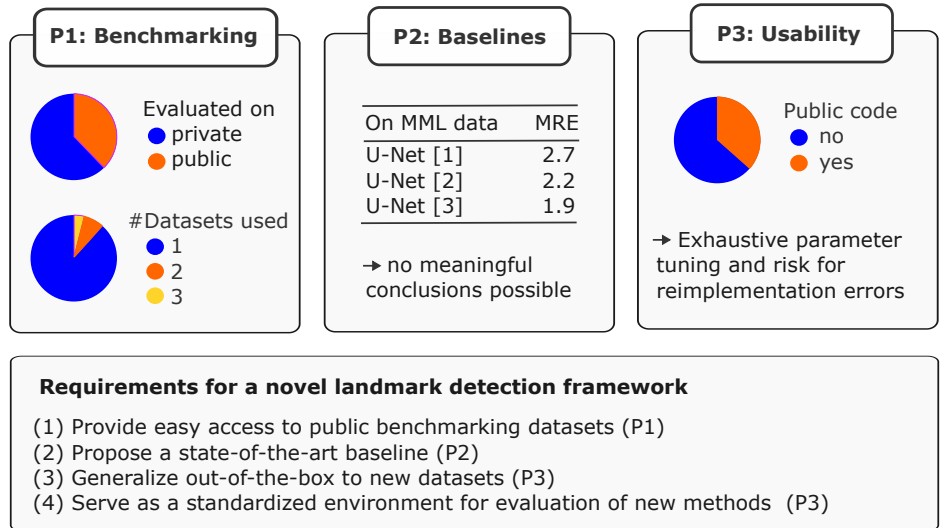

Figure 1: We identified three key pitfalls in the current landmark detection literature regarding benchmarking, baseline comparison and usability, and formulated four practical requirements for a new framework tackling these shortcomings. [1] (Zhang et al., 2024) [2] (He et al., 2024) [3] (Huang et al., 2025)

.

## 2. Method

Tackling the current pitfalls in landmark detection we derive four practical requirements for a newly proposed framework: (1) Provide easy access to public benchmarking datasets; (2) propose a strong, common baseline, achieving state-of-the-art accuracy across datasets; (3) provide out-of-the-box generalizability for training on new datasets without the need for manual intervention; (4) serve as a flexible and standardized environment for evaluation of new methodological developments. In segmentation, these requirements have long been understood and are addressed by the well-established nnU-Net framework, which consistently delivers state-of-the-art performance across various datasets (Isensee et al., 2021, 2024). The key concept of nnU-Net is its self-configuration to the task at hand by automatically deriving dataset-specific properties and adjusting preprocessing and hyperparameters for a (residual) U-Net architecture. Further, nnU-Net has implemented many best practices of image processing for example regarding its data augmentation pipeline and sliding window prediction, which allow translation to heatmap regression. It is therefore well-motivated to reuse nnU-Net as a self-configuration and training engine. In the following we explore how nnLandmark can be built on this existing infrastructure to arrive at a state-of-the-art, generalizable framework for landmark detection.

To leverage nnU-Net's data loading pipeline, which expects segmentation inputs, we initially store the landmarks in a multi-label segmentation map, with each landmark represented by a $3 \times 3 \times 3$ voxel label. This way nnLandmark can exploit the fully automatic preprocessing and self-configuration machinery, which has been extensively tuned for 3D medical segmen-

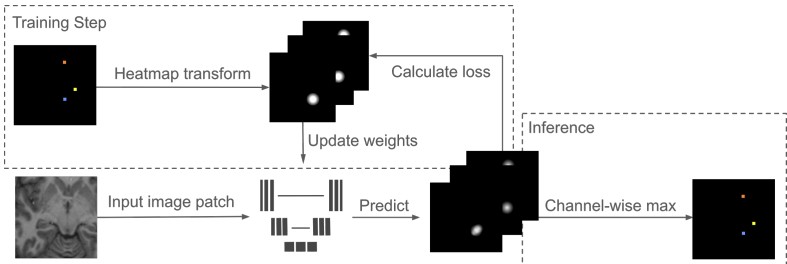

Figure 2: Leveraging nnU-Net's data loading and augmentation pipeline, landmark seg-
mentation maps are transformed to heatmaps only during loss computation, each
landmark represented by a EDT in a dedicated channel. In inference, landmark
coordinates are identified by the channel-wise maximum.

tation. The conversion from this multi-label representation to heatmap regression happens
after data augmentation, directly inside the loss computation. For each foreground label,
i.e. landmark, the target coordinate is obtained as the center of mass of the corresponding
label region. Around this point, an Euclidean distance transform (EDT) with a radius of
15 voxels is injected into a dedicated output channel as the regression target, yielding a
smooth distance-based heatmap (Figure 2). This on-the-fly transformation further avoids
memory- and CPU-intensive storing and loading of large heatmaps. During prediction, a
sigmoid activation in the final layer constrains voxel intensities to [0,1], stabilizing training.
Heatmap regression is trained with a Binary Cross-Entropy (BCE) TopK20 loss, which fo-
cuses the gradient signal on the most challenging voxels. Concretely, for every patch the
voxel-wise BCE values are ranked and only the voxels with the highest 20% loss values con-
tribute to the final loss, which can mitigate the foreground-background imbalance inherent
to sparse landmark heatmaps. For inference, we adapt nnU-Net's sliding window prediction
and derive landmark coordinates by taking the channel-wise maximum of the heatmap.

## 3. Experiments

### 3.1. Metrics

The **Mean Radial Error (MRE)** measures the average Euclidean distance between the
predicted and ground truth landmark coordinates. It is defined as:

$$\text{MRE} = \frac{1}{N} \sum_{i=1}^{N} ||\mathbf{x} - \hat{\mathbf{x}}||_2 \tag{1}$$

where $\mathbf{x}$ and $\hat{\mathbf{x}}$ represent the ground truth and predicted coordinates for the $i$-th landmark,
respectively, and $N$ is the total number of landmarks. Lower MRE values indicate higher
localization accuracy.

The **Success Detection Rate (SDR)** within a tolerance range quantifies the proportion
of detected landmarks that fall within a specified distance threshold $t$ from their ground

truth positions, defined as:

$$\text{SDR@t} = \frac{\#\text{ landmarks with MRE} \leq \text{t}}{\#\text{ landmarks}} \times 100. \tag{2}$$

## 3.2. Datasets

We evaluate three public and one private dataset spanning various 3D imaging modalities and anatomical regions. All test splits were used as hold-out test data.

The **Mandibular Molar Landmarking (MML)** dataset (He et al., 2024) provides 648 CT images along with annotations of 14 dental landmarks targeting the crowns and roots of the second and third mandibular molars. The dataset comes with the challenge of missing landmarks in cases where teeth are absent, structurally damaged, or have irregular root anatomy. However, we focus on predicting complete landmark annotations and use a subset that only included fully annotated cases, hereafter referred to as the complete MML subset. In accordance with the official split, the complete subset contains 283 training, 56 validation, and 60 test cases. We used the official train/test split; the validation split was not used in this study.

The **Anatomical Fiducials (AFIDs)** dataset (Taha et al., 2023; Abbass et al., 2022) consists of 132 T1 brain MRI images with 32 annotated brain landmarks as anatomical fiducials. AFIDs is a collection of 4 subsets: (1) the AFIDs-HCP30 dataset (n=30), 3T scans from the Human Connectome Project (HCP) (https://ida.loni.usc.edu/login.jsp) (Van Essen et al., 2012); (2) the AFIDs-OASIS30 dataset (n=30), 3T scans from the Open Access Series of Imaging Studies OASIS-1 (Marcus et al., 2007); (3) the London Health Sciences Center Parkinson's disease (LHSCPD) dataset (n=40) containing gadolinium-enhanced images from a 1.5 T scanner (Abbass et al., 2023), and (4) the Stereotactic Neurosurgery (SNSX) (Lau et al., 2023) dataset (n=32) acquired with a 7T head-only scanner. Thus, this dataset is highly heterogeneous, with subdatasets differing in origin and imaging protocols. The human error on this dataset is reported as 0.99 mm with an inter-rater variability of 1.48 mm. As there is no official split, we performed a random split stratified across the four subsets into 110 training and 22 test cases, which will be published with the code.

The **fetal pose estimation** dataset (Chen et al., 2020, 2024) is a private dataset provided by Shenzhen University. It encompasses 1000 fetal ultrasound (US) images with 22 landmarks throughout the head, trunk and limbs, allowing to monitor fetal position and development.

The **Fetal Cerebellum Landmark Detection (LFC)** dataset (Gong et al., 2025c,b,a) contains fetal brain T2 MRIs. The annotations target 12 landmarks, which represent control points for 6 biometric measurements concerning the cerebellum, brain and skull diameters. We use the official train/test split of 120/60. For LFC, we evaluated both landmark detection and the resulting measurements, as the dataset primarily targets the downstream task of fetal brain biometry rather than precise landmark placement. Further, some measurements can be taken in slightly shifted locations while still producing accurate values.

## 3.3. Benchmarking and Training Details

For nnLandmark, preprocessing and hyperparameters, such as patch size, batch size, network topology, are configured automatically by the framework based on dataset-specific

properties. All nnLandmark experiments were performed using the *3d_fullres* preset and 5-fold cross-validation. We trained with the plain U-Net architecture, as well as variations with a ResNet-based encoder in two sizes, M and L (ResEncM/L), following official nnU-Net recommendations (Isensee et al., 2024). We compared nnLandmark to three recently published and state-of-the-art methods and toolkits. All baselines were trained on the four datasets, utilizing the official code and recommendations. The goal is to compare entire frameworks and repositories against each other to evaluate their generalizability to new datasets, without requiring any custom changes or elaborate hyperparameter tuning. We additionally integrated the H3DE architecture (Huang et al., 2025) into nnLandmark to demonstrate its utility as a powerful, standardized method development framework.

The **Hybrid-3D Network (H3DE-Net)** (Huang et al., 2025) integrates transformer-based attention modules within a U-Net-like CNN structure to effectively handle local feature extraction as well as global context modeling. The design of downsampling layers and window configuration restricts the input shape to be divisible by $64 \times 64 \times 32$. MML training was done using a random cropping data augmentation to $128 \times 128 \times 64$ voxels, to also fit the test image shape. For the remaining experiments, images were resized to $128^3$ voxels. **Landmarker** (Jonkers et al., 2025b,a) is a toolkit which offers useful modules for handling landmark data and frequently used architectures in the domain. They provide a set of default configurations that show promising results on the MML data (Jonkers et al., 2025a), using a Flexible U-Net with EfficientNet backbone. Following the practices of the authors, for MML, the training data were cropped based on the annotations to $128 \times 128 \times 64$ voxels, to fit the field of view in the test data. For the remaining datasets, images were resized to $128^3$. The models were trained as an ensemble using five different seeds. The **Super-Resolution U-Net (SR-UNet)** (Zhang et al., 2024) adopts pyramid pooling and super-resolution blocks to better preserve details and mitigate the error caused by downsampling and upsampling operations during training. All data was preprocessed using the published heatmap conversion script, which also includes resizing to $128^3$. For MML they report cropping the train data to $128 \times 128 \times 64$ voxels, so we use the same label-based crops as for Landmarker.

## 4. Results

nnLandmark, particularly in the ResEncM configuration, demonstrated the overall highest performance. For the MML dataset, reproducibility results are reported in Appendix D. For H3DE we only saw slight deviation from the reported results, attributable to some randomness during training (Huang et al., 2025). For SR-UNet and Landmarker however results could not be reproduced, with both yielding an MRE above 10 mm, despite using the official repositories. This might be due to differences in handling the shift in field of view from whole-head CT during training to already cropped images in the test set. nnLandmark handles this inherently due to its patch-wise training scheme and sliding window prediction, and H3DE added random cropping during training. Landmarker required cropping the training images based on the labels to resemble the test shape. SR-UNet similarly reports random cropping of the images. Further, the reported Landmarker results were obtained on a custom randomized data split after cropping (Jonkers et al., 2025a), hindering comparability. On AFIDs, all baselines showed moderate MRE of 3 mm to 4 mm. nnLandmark ResEncM achieved an MRE of 1.46 mm, falling within the reported inter-rater variability of 1.48 mm

Table 1: Results of landmark detection performance of nnLandmark compared to current state-of-the-art methods on hold-out test data of four datasets, evaluated by MRE and micro standard deviation (std). All models were trained using the official code.

| Method | MRE±Std [mm] | | | |
|---|---|---|---|---|
| | MML | AFIDs | Fetal pose | LFC |
| H3DE | 1.81±1.15 | 4.28±2.09 | 6.07±6.44 | 4.22±4.21 |
| SR-UNet | 10.01±10.37 | 3.37±1.97 | 66.11±19.15 | 3.92±3.62 |
| landmarker | 10.58±13.92 | 2.86±4.12 | 5.37±7.99 | 4.02±5.13 |
| nnLandmark w/ H3DE | 1.63±1.16 | 1.79±1.05 | 4.25±6.33 | **3.72±4.47** |
| nnLandmark | 1.39±0.85 | 1.55±1.01 | 3.15±5.01 | 3.82±4.67 |
| nnLandmark ResEncM | **1.36±0.88** | **1.46±1.01** | 3.06±4.51 | 3.75±4.77 |
| nnLandmark ResEncL | 1.59±1.22 | 1.61±1.06 | **3.05±4.52** | 3.75±4.75 |

Table 2: Results of biometry measurements on the LFC dataset.

| Method | Measurement error ± Std [mm] |
|---|---|
| H3DE (Huang et al., 2025) | 2.85±2.32 |
| SR-UNet (Zhang et al., 2024) | 5.92±5.55 |
| landmarker (Jonkers et al., 2025b) | 1.78±4.25 |
| nnLandmark w/ H3DE | 1.52±1.24 |
| nnLandmark | 1.79±1.94 |
| nnLandmark ResEncM | 1.20±0.93 |
| **nnLandmark ResEncL** | **1.17±0.94** |

(Taha et al., 2023). For the fetal pose estimation task, H3DE and Landmarker showed a moderate MRE of 5 mm to 6 mm, while SR-UNet did not converge. All models show a high standard deviation on this dataset, reflecting the challenging nature of the task, where the fetus can have varying positions in the uterus, arms and legs can be crossed and the respective landmarks consequently easily confused. On LFC, all models show a moderate MRE of about 4 mm and comparably high standard deviation, which could be attributed to the shifted annotation of some control points. A large error in landmark localization can still lead to accurate biometry measurements, which is also reflected in our results.

Integrating the H3DE architecture in our nnLandmark framework strongly improved accuracy compared to using H3DE with the published repository on all datasets. While it even achieved a slight advantage in LFC landmark localization compared to nnLandmark U-Nets, this didn't show in the biometry measurement results. For all datasets, MRE for each landmark class individually are presented in Appendix C. Randomly chosen example predictions for the nnLandmark ResEncM model for all datasets are shown in Figure 3.

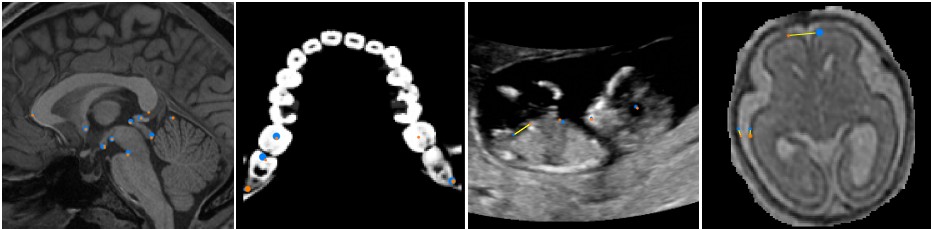

Figure 3: Qualitative results for each test subset with ground truth (blue), model prediction (orange) and error (yellow). If the landmarks are actually present in the neighbouring slice of the shown one, the points are reduced in size.

## 5. Discussion

Current research in 3D landmark detection lacks the foundation needed for systematic progress, including transparent benchmarking, consistent baselines, and methods that reliably generalize across datasets (Figure 1). Consequently, new methods are often not evaluated in a broader, standardized context and their translation to new datasets can require substantial manual effort, leading to a gap of more general solutions and insights. Tackling these pitfalls, we introduce nnLandmark, the first self-configuring framework for 3D medical landmark detection. Leveraging the established infrastructure of nnU-Net we inherit extensively optimized components for preprocessing, data augmentation and training while extending the framework with a dedicated heatmap representation, adapted loss computation, and coordinate prediction logic. This combination creates the first solution in the field to automatically adapt to new datasets without the need for expert intervention. Thereby, nnLandmark occupies the unique position of serving as an out-of-the-box usable baseline as well as a flexible framework for method development and standardized evaluation, enabling transparent and comparable experimentation in the field.

To ensure a comprehensive evaluation, we assessed nnLandmark on three public and one private dataset spanning different modalities and anatomical regions and compare against three recently published methods and frameworks, H3DE, SR-UNet and Landmarker (Huang et al., 2025; Zhang et al., 2024; Jonkers et al., 2025b). Although all three report strong performance on MML, systematic evaluation beyond this dataset has been missing. Our benchmarking closes this gap and highlights the need for broader evaluation as a standard practice in medical landmark detection. The scarcity of established benchmarks and the common reliance on single, often private datasets have limited the comparability of methods. Our results show that performance shifts considerably across datasets, underlining the importance of designing and validating methods with generalization in mind. To facilitate broader adoption of multi-dataset evaluation, we release data conversion scripts for relevant public benchmarks, enabling straightforward use within the nnLandmark framework.

On the MML dataset, we also observed substantial variability among 3D U-Net baselines reported in the literature, with published MRE values ranging from 1.90 mm to 2.70 mm despite nominally identical architectures (He et al., 2024; Huang et al., 2025; Zhang et al., 2024). These discrepancies demonstrate that architectural design alone is insufficient; optimal preprocessing, hyperparameter configuration, and training practices are equally im-

portant to achieve reliable performance. nnLandmark addresses these issues as its self-configuring design eliminates the need for manual adjustments, providing a standardized, high-performing baseline for 3D landmark detection. nnLandmark's automatic adaptation to the dataset at hand further makes it the first framework to allow training state-of-the-art landmark detection models on new datasets without the need for expert knowledge and manual tuning.

Finally, the framework provides a controlled environment for developing and ablating new methodological ideas, relieving researchers from the need to build custom experimentation code while substantially improving the comparability of results. We illustrate this by integrating the H3DE architecture into nnLandmark, which further achieved improved performance compared to the official repository. These findings highlight the importance of a well-configured, standardized experimental environment for drawing meaningful, broadly applicable conclusions and reliably assessing methodological progress (Isensee et al., 2024). While nnLandmark addresses several long-standing challenges in landmark detection, some limitations remain. A limitation of storing labels as multi-label segmentation maps is the handling of closely spaced landmarks. Since each landmark is encoded as a $3 \times 3 \times 3$ voxel cube, two landmarks must be separated by at least three voxels to avoid overlap that would distort their encoded locations. In addition, nnLandmark's current inference design always predicts a complete set of landmarks by taking the argmax of each heatmap channel. This does not account for anatomically absent landmarks, which occur, for example, in the full MML dataset where teeth may be missing. Handling presence or absence could be incorporated by estimating a confidence threshold from cross-validation and suppressing predictions below that threshold. The same mechanism could extend the framework to small object detection, where thresholding would allow multiple instances per class. Implementing reliable presence detection and multi-instance prediction is left for future work.

## 6. Conclusion

nnLandmark is introduced as a self-configuring deep learning framework for 3D medical landmark localization based on heatmap regression. It addresses three key pitfalls of the current literature: the lack of public benchmarking, inconsistent baseline implementations, and limited out-of-the-box usability. By conducting a benchmarking study across three public and one private dataset, nnLandmark establishes a transparent reference for evaluating existing and future methods. Building on established components from nnU-Net, the framework translates self-configuration concepts to landmark detection, while providing a tailored heatmap generation, loss design, and inference logic. This enables robust generalization to new datasets without task-specific hyperparameter tuning or expert intervention. At the same time, nnLandmark offers a standardized, ready-to-use baseline and a flexible environment for method development, supporting reproducible experiments and systematic ablations. Together, these properties lay the groundwork for more rigorous and comparable research in 3D medical landmark detection, where novel ideas can be evaluated transparently and genuine methodological progress becomes measurable.

## Acknowledgments

Regarding the AFIDs-HCP dataset: Data collection and sharing for this project was provided by the Human Connectome Project (HCP; Principal Investigators: Bruce Rosen, M.D., Ph.D., Arthur W. Toga, Ph.D., Van J. Weeden, MD). HCP funding was provided by the National Institute of Dental and Craniofacial Research (NIDCR), the National Institute of Mental Health (NIMH), and the National Institute of Neurological Disorders and Stroke (NINDS). HCP data are disseminated by the Laboratory of Neuro Imaging at the University of Southern California. Regarding the AFIDs-OASIS dataset: Data were provided by OASIS-1: Cross-Sectional: Principal Investigators: D. Marcus, R, Buckner, J, Csernansky J. Morris; P50 AG05681, P01 AG03991, P01 AG026276, R01 AG021910, P20 MH071616, U24 RR021382. Part of this work was funded by Helmholtz Imaging (HI), a platform of the Helmholtz Incubator on Information and Data Science.

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

## Appendix A. List of analyzed publications for Figure 1

We identified current pitfalls in the 3D medical landmark detection literature based on the following representative list of relevant methodological publications since 2021: (Huang et al., 2025; Zhang et al., 2024; Jonkers et al., 2025b; Gong et al., 2025a; He et al., 2024; Chen et al., 2021, 2024; Cui et al., 2025; Liu et al., 2023; Pang et al., 2024; Gong et al., 2025b; Barough et al., 2025; Chen et al., 2022; Kang et al., 2021; Jiang et al., 2022; Shi et al., 2025; Dai et al., 2024; Stebani et al., 2023; Baksi et al., 2021; Gillot et al., 2023; Salari et al., 2023; Wu et al., 2022; Li et al., 2023; Lu et al., 2023; Lang et al., 2022b; López Diez et al., 2021; Zhu et al., 2022; Lang et al., 2022a; Dot et al., 2022; Torosdagli et al., 2023)

# Appendix B. Extended Results Including Success Detection Rate (SDR)

Table 3: Extended results table with SDR. All results are done on the hold-out testsplits.

| Method | MRE±Std [mm] | SDR [%] | | |
|---|---|---|---|---|
| | | 2 mm | 3 mm | 4 mm |
| **AFIDs** #samples 22, #landmarks 32 | | | | |
| H3DE (Huang et al., 2025) | 4.28±2.09 | 13.07 | 28.98 | 51.28 |
| SR-UNet (Zhang et al., 2024) | 3.37±1.97 | 25.43 | 50.99 | 70.03 |
| landmarker (Jonkers et al., 2025b) | 2.86±4.12 | 46.16 | 62.22 | 74.57 |
| nnLandmark w/ H3DE | 1.79±1.05 | 67.90 | 88.78 | 97.02 |
| nnLandmark | 1.55±1.01 | 76.85 | 93.61 | 97.87 |
| **nnLandmark ResEncM** | **1.46±1.01** | **81.82** | **94.74** | **98.15** |
| nnLandmark ResEncL | 1.61±1.06 | 75.71 | 92.90 | 97.44 |
| **MML complete subset** #samples 60, #landmarks 14 | | | | |
| H3DE (Huang et al., 2025) | 1.81±1.15 | 67.14 | 89.52 | 97.14 |
| SR-UNet (Zhang et al., 2024) | 10.01±10.37 | 5.24 | 13.93 | 24.17 |
| landmarker (Jonkers et al., 2025b) | 10.58±13.92 | 24.05 | 34.05 | 42.62 |
| nnLandmark w/ H3DE | 1.63±1.16 | 72.50 | 91.19 | 97.26 |
| nnLandmark | 1.39±0.85 | 80.00 | 95.24 | 98.57 |
| **nnLandmark ResEncM** | **1.36±0.88** | **82.02** | **95.48** | **98.69** |
| nnLandmark ResEncL | 1.59±1.22 | 75.24 | 92.50 | 97.98 |
| **Fetal pose** #samples 200, #landmarks 22 | | | | |
| H3DE (Huang et al., 2025) | 6.07±6.44 | 11.21 | 27.81 | 45.02 |
| SR-UNet (Zhang et al., 2024) | 66.11±19.15 | 0.04 | 0.04 | 0.04 |
| landmarker (Jonkers et al., 2025b) | 5.37±7.99 | 36.45 | 56.16 | 67.36 |
| nnLandmark w/ H3DE | 4.35±6.33 | 44.77 | 64.23 | 74.77 |
| nnLandmark | 3.15±5.01 | 49.77 | 70.86 | 82.16 |
| nnLandmark ResEncM | 3.06±4.51 | 50.80 | 70.64 | 81.68 |
| **nnLandmark ResEncL** | **3.05±4.52** | **51.19** | **71.09** | **81.80** |
| **LFC** #samples 60, #landmarks 12 | | | | |
| H3DE (Huang et al., 2025) | 4.22±4.21 | 34.03 | 55.42 | 67.78 |
| SR-UNet (Zhang et al., 2024) | 3.92±3.62 | 35.42 | 55.97 | 69.72 |
| landmarker (Jonkers et al., 2025b) | 4.02±5.13 | 50.56 | 68.19 | 76.67 |
| nnLandmark w/ H3DE | 1.63±1.16 | 72.50 | 91.19 | 97.26 |
| nnLandmark | 3.72±4.47 | 51.94 | 67.91 | 77.22 |
| **nnLandmark ResEncM** | **3.75±4.77** | **55.42** | **69.17** | **77.08** |
| nnLandmark ResEncL | 3.75±4.75 | 54.44 | 68.61 | 76.53 |

## Appendix C. Individual Landmark Class Errors

Table 4: Landmark localization results of nnLandmark ResEncM for AFIDs dataset with #samples 22, #landmarks 32

| Landmark Class | MRE±Std |
| --- | --- |
| AC [midline] | 0.68±0.34 |
| PC [midline] | 0.90±0.39 |
| Infracollicular sulcus [midline] | 1.19±0.39 |
| Pontomesencephalic junction [midline] | 1.38±0.73 |
| Superior interpeduncular fossa [midline] | 0.87±0.35 |
| Right superior lateral mesencephalic sulcus | 1.19±0.50 |
| Left superior lateral mesencephalic sulcus | 1.06±0.54 |
| Right inferior lateral mesencephalic sulcus | 1.43±0.67 |
| Left inferior lateral mesencephalic sulcus | 1.35±0.68 |
| Culmen [midline] | 1.79±0.95 |
| Intermammillary sulcus [midline] | 1.10±0.46 |
| Right mammillary body | 0.95±0.42 |
| Left mamillary body | 1.12±0.50 |
| Pineal gland [midline] | 1.61±0.84 |
| Right lateral aspect of frontal horn at AC | 1.70±1.13 |
| Left lateral aspect of frontal horn at AC | 1.95±1.19 |
| Right lateral aspect of frontal horn at PC | 1.86±0.92 |
| Left lateral aspect of frontal horn at PC | 1.71±0.93 |
| Genu of corpus callosum [midline] | 1.10±0.39 |
| Splenium of the corpus callosum [midline] | 1.22±0.40 |
| Right anterolateral temporal horn | 1.16±0.70 |
| Left anterolateral temporal horn | 1.45±0.52 |
| Right superior AM temporal horn | 1.45±0.62 |
| Left superior AM temporal horn | 1.95±0.93 |
| Right inferior AM temporal horn | 2.14±0.91 |
| Left inferior AM temporal horn | 2.20±1.08 |
| Right indusium griseum origin | 1.42±0.78 |
| Left indusium griseum origin | 1.75±0.64 |
| Right ventral occipital horn | 1.80±1.60 |
| Left ventral occipital horn | 2.46±3.03 |
| Right olfactory sulcal fundus | 1.43±0.65 |
| Left olfactory sulcal fundus | 1.20±0.45 |

Table 5: Landmark localization results of nnLandmark ResEncM for MML dataset with #samples 60, #landmarks 14

| Landmark Class | MRE±Std |
|---|---|
| Left cuspid cusp | 1.29±1.06 |
| Left 2nd molar crown | 0.96±0.56 |
| Left 2nd molar mesial root | 1.26±0.65 |
| Left 2nd molar distal root | 1.33±0.75 |
| Left 3rd molar crown | 1.20±0.68 |
| Left 3rd molar mesial root | 1.63±1.31 |
| Left 3rd molar distal root | 1.90±0.89 |
| Right cuspid cusp | 1.29±0.77 |
| Right 2nd molar crown | 1.17±0.61 |
| Right 2nd molar mesial root | 1.25±0.61 |
| Right 2nd molar distal root | 1.41±0.79 |
| Right 3rd molar crown | 1.04±0.46 |
| Right 3rd molar mesial root | 1.50±0.81 |
| Right 3rd molar distal root | 1.74±1.28 |

Table 6: Landmark localization results of nnLandmark ResEncM for fetal pose estimation dataset with #samples 200, #landmarks 22

| Landmark Class | MRE±Std |
|---|---|
| Cranial crest | 5.15±8.43 |
| Diencephalon | 2.56±4.74 |
| Thalamus | 1.84±1.32 |
| Nasal bone | 1.93±2.57 |
| Lower alveolar | 1.47±1.09 |
| Hind neck | 3.78±3.70 |
| Chest wall | 3.85±3.04 |
| Diaphragm lumbar | 3.71±4.10 |
| Buttocks | 3.98±5.91 |
| Umbilical | 4.09±4.41 |
| Left shoulder | 2.15±1.99 |
| Left elbow | 2.37±3.35 |
| Left wrist | 3.00±4.86 |
| Right shoulder | 2.99±5.39 |
| Right elbow | 2.54±4.00 |
| Right wrist | 3.13±4.95 |
| Left hip | 2.37±3.90 |
| Left knee | 2.60±4.10 |
| Left ankle | 3.96±5.24 |
| Right hip | 2.75±5.22 |
| Right knee | 2.55±2.99 |
| Right ankle | 4.51±5.26 |

Table 7: Landmark localization results of nnLandmark ResEncM for LFC dataset with #samples 60, #landmarks 12

| Landmark Class | MRE±Std |
|---|---|
| Brain biparietal diameter 1 (bBIP1) | 6.09±5.76 |
| Brain biparietal diameter 2 (bBIP2) | 6.15±5.74 |
| Skull biparietal diameter 1 (sBIP1) | 6.11±4.77 |
| Skull biparietal diameter 2 (sBIP2) | 5.99±4.97 |
| Transverse cerebellar diameter 1 (TCD1) | 1.30±0.64 |
| Transverse cerebellar diameter 2 (TCD2) | 1.21±0.69 |
| Occipitofrontal diameter 1 (OFD1) | 6.52±7.32 |
| Occipitofrontal diameter 2 (OFD2) | 6.09±5.83 |
| Height of vermis 1 (HDV1) | 1.41±0.66 |
| Height of vermis 2 (HDV2) | 1.45±0.58 |
| Anteroposterior diameter of vermis 1 (ADV1) | 1.57±0.68 |
| Anteroposterior diameter of vermis 2 (ADV2) | 1.14±0.55 |

Table 8: Landmark localization results of nnLandmark ResEncM for LFC biometry measurement with #samples 60, #measurements 6

| Landmark Class | MRE±Std |
|---|---|
| Brain biparietal diameter (bBIP) | 1.49±1.23 |
| Skull biparietal diameter (sBIP) | 1.24±0.94 |
| Transverse cerebellar diameter (TCD) | 1.07±0.76 |
| Occipitofrontal diameter (OFD) | 1.25±0.94 |
| Height of vermis (HDV) | 1.30±0.84 |
| Anteroposterior diameter of vermis (ADV) | 0.85±0.65 |

# Appendix D. Reproducibility results

Table 9: Reproducibility results on the complete subset of the Mandibular Molar Landmark (MML) dataset.

| Method | MRE±Std [mm] |
|---|---|
| H3DE reported (Huang et al., 2025) | 1.68±0.45 |
| H3DE reproduced (Huang et al., 2025) | 1.81±1.15 |
| SR-UNet reported (Zhang et al., 2024) | 2.01±4.33 |
| SR-UNet reproduced (Zhang et al., 2024) | 10.01±10.37 |
| landmarker reported (different split) (Jonkers et al., 2025b) | 1.39 |
| landmarker reproduced (Jonkers et al., 2025b) | 10.58±13.92 |

