# OpenReview forum: "nnLandmark: A Self-Configuring Method for 3D Medical Landmark Detection"
_MIDL.io/2026/Conference — MIDL 2026 Poster_

### Official Review · Reviewer_1sG2 · 2026-01-02

**Confidence:** 5
**Preliminary Rating:** 5
**Final Rating:** 5

**Summary:**

The author proposes changes to nnUnet to change it to a Landmark model. They argue about the lack of reproducibility of code, the availability of public data sets, and the difficulty of guaranteeing a fair hyperparameter search problem for baselines. They claim that with the nnUnet hyperparameter, they can guarantee that a decent no-hyperparameter search baseline is possible.

**Strengths:**

If the setup is equal to the nnUnet, it would produce an easy Benchmark and/or utility for Application only.

The paper discusses the current lack of a baseline in this field.

Multiple Benchmarks with many different image types and targets.

**Weaknesses:**

The model uses voxel space to encode their points, which will cause rounding errors and make subvoxel measurement impossible with this method.

The claim that the nnUnet h-parameter is optimal is not tested. (In the paper, there is no strong claim that this is the case, only implicit)

**Detailed Comments:**

-

**Justification Of Final Rating:**

Strong baselines and results between different modalities and targets.
Important for reproducibility benchmarks and pure applications.

The paper shows that a good segmentation model can be transformed into a Segmentation model.

**Justification Of The Preliminary Rating:**

Strong baselines and results between different modalities and targets.
Important for reproducibility benchmarks and pure applications.

The paper shows that a good segmentation model can be transformed into a Segmentation model.

**Questions To Address In The Rebuttal:**

Why did you choose a pixel-based input instead of a coordinate system?

In my experience, the resolution is strongly related to the error in point prediction, as indicated by heat maps. Did you observe the same? Should there be a discussion with the user to allow them to choose between the fast/small/low-res and the slow/big/high-res models, depending on their needs?

---

> ### Author Response · Authors · 2026-01-24
> **Pixel-based Input, Target Spacing and Model Variants**
>
> Many thanks to all reviewers for the constructive feedback and interesting questions.
>
> - Pixel-based input. Our current infrastructure for data augmentation is defined as a dense resampling operation. This way, labels can be transformed by the exact same operations as the image ensuring consistency under interpolation, discretization, padding, and boundary handling. In contrast, directly transforming coordinate-valued landmarks would require either restricting augmentations to simple affine transforms or approximating the inverse of a dense deformation field, both of which introduce avoidable inaccuracies.
>
> - Model variants. nnLandmark allows the user to choose from the same configuration as nnU-Net, this includes 3D highres (default recommendation), 3D lowres and a 3D lowres-highres cascade. Further the framework has access to the same network designs, including the standard U-Net and ResEnc M, L and XL variants, employing a ResNet encoder [1]. In segmentation, these architectures have shown improved performance especially on large datasets. While the overhead of ResEncM is moderate, the L variants require significantly longer runtime and VRAM. Our results so far indicate no advantage of the larger ResEncL and our default recommendation for the best results is the ResEncM.
>
> - Target resolution. The network target resolution is selected automatically during the preprocessing and experiment planning pipeline. The target spacing to the per-axis median spacing of the training data. This reflects the usual context–precision trade-off (lower resolution yields more context; higher resolution yields more precision); we thereby adopt nnU-Net’s proven default, but users can override the target spacing in the experiment planning as desired.
>
> - Sub-voxel accuracy. Currently we don’t have a mechanism implemented for sub-voxel accuracy. If the landmarks were annotated in a higher resolution, a workaround could be to upsample the images to the annotation resolution. Further, heatmaps generally allow sub-pixel precision by adding gaussian fitting into the postprocessing and reporting the center location, instead of the current default which is taking the channel-wise argmax. The gaussian fitting could be integrated quite straightforwardly into nnLandmark’s inference class.
>
> - Hyperparameter ablation. We added ablation studies on the robustness of our hyperparameter choice regarding landmark specific parameters including the size of the EDT for representing the landmarks in the heatmaps and the design of the loss function (Appendix B).
>
>
> We further would like to note that we added two additional datasets to our evaluation, the FeTA biometry and PDDCA dataset, to additionally strengthen our contribution.
>
>
> [1] https://github.com/MIC-DKFZ/nnUNet/blob/master/documentation/resenc_presets.md

---

### Official Review · Reviewer_uEFD · 2026-01-09

**Confidence:** 4
**Preliminary Rating:** 3
**Final Rating:** 4

**Summary:**

This paper presents nnLandmark, which is a self-configuring framework for 3D medical landmark detection. The framework is motivated by the nnUNet, and aims to address the current limitations, including limited public benchmarking, inconsistent baseline implementation, and limited out-of-the-box usability.

**Strengths:**

1. The paper is well-motivated by nnUNet and the proposed nnLandmark can serve as a strong and standard baseline for 3D medical landmark detection.
2. The experiment that integrates H3DE into nnLandmark has shown the advantage of a standardized preprocessing and training pipeline.
3. The results have demonstrated nnLandmark’s consistent superior performance across multiple datasets and imaging modalities.

**Weaknesses:**

1. Some ablation study may be missed in terms of the loss and hyperparameter selection, for example, the number of Top k. Another option is to present a reasonable explanation on the choice.
2. The visual results seem to be insufficient, including more cases and predictions from other methods can be useful for limitation understanding.
3. The novelty of the methods remains a concern, and it is primarily at engineering level.

**Detailed Comments:**

See Strengths and Weaknesses

**Justification Of Final Rating:**

I really appreciate author's response and additional experiments on ablation study. Including two additional datasets makes the paper more convincing. Therefore, I am happy to raise my score and look forward to the codebase.

**Justification Of The Preliminary Rating:**

Although there is a concern on the method’s novelty due to nnUNet, the paper can serve as a standard baseline for 3D medical landmark detection. However, publishing the codebase will be necessary given the motivation of nnLandmark.

**Questions To Address In The Rebuttal:**

Including ablation study and more visual results. Publish the codebase.

---

> ### Author Response · Authors · 2026-01-24
> **Visual Examples, Ablations and Novelty**
>
> Many thanks to all reviewers for the constructive feedback and interesting questions.
>
> - Ablation study and visual results. We added ablation studies on the robustness of our hyperparameter choice regarding landmark specific parameters including the size of the EDT for representing the landmarks in the heatmaps and the design of the loss function (Appendix B). We further added comprehensive qualitative examples (Appendix F) for all methods and all datasets.
>
> - Novelty. We agree that we rely on established components, like employing a U-Net architecture for heatmap regression. However, following Michael Black’s guide [1], novelty can also show in changes of existing ideas which result in significant improvements. We demonstrate how to optimally apply these established techniques within a novel framework and show that a properly configured 3D U-Net can produce state-of-the-art results. This is a novel and impactful insight for the landmark detection community, challenging a prevalent misconception that 3D U-Nets perform insufficient as they are generally presented as rather subpar baselines in the current literature.
> nnLandmark is the first self-configuring framework for 3D landmark detection, which works reliably on a large range of datasets and finally enables access to automated landmark detection with no need for expert knowledge.
> Previously progress was hampered through subpar baselines and a lack of standardized frameworks to evaluate new ideas. nnLandmark is the first of its kind to deliver exactly that, while delivering state-of-the-art performance. In parallel to how nnU-Net has revolutionized 3D semantic segmentation, nnLandmark is set to become the new default method and catalyst for scientific innovation.
>
>
> We further would like to note that we added two additional datasets to our evaluation, the FeTA biometry and PDDCA dataset, to additionally strengthen our contribution.
>
>
> [1] https://medium.com/@black_51980/novelty-in-science-8f1fd1a0a143

---

### Official Review · Reviewer_CDFo · 2026-01-12

**Confidence:** 3
**Preliminary Rating:** 4
**Final Rating:** 5

**Summary:**

nnLandmark provides out-of-the-box benchmarked baseline framework for medical image landmarking tasks on 3 public datasets and 1 private dataset compared against 3 representative methods whose codebase are available. It is based on the popular nnUnet framework but adapted to anatomical landmarks.

To adapt nnUnet to Heatmap regression based landmark detection task, the gaussian blobs are generated on-the-fly and corresponding BCE-TopK20 loss is used. Performance of the proposed baseline against compared methods show improved results on the landmark prediction metrics.

**Strengths:**

The adaptation of widely validated nnUnet framework via adapting the nnUnet-based segmentation pipeline to heatmap regression-based landmark task is a plus. The contributions stated by the paper are backed by the experiment and well motivated. The problem of reproducibility and out-of-the-box usability is demonstrated well.
Additionally, the writing is easy to follow.

**Weaknesses:**

*Somewhat limited method benchmarking* : I understand that the paper’s focus is on the out-of-the-box framework for landmark regression and show that it is a good baseline for wide landmarking datasets. The benchmarking of architectures is mostly focused on dense prediction based ones, specifically - heatmap regression. But a comparison to show its competitiveness to alternatives like coordinate regression and its variants such as predicting offsets from known points (ABRM-He et.al 2024).

**Detailed Comments:**

To what extent is the nnUnet framework adapted to nnLandmark? For example: Is a two-stage pipeline (coarse-to-fine) also implemented in nnLandmark similar to nnUnet? Any results comparing one-stage to two-stage pipeline, if available?

*Discussion of Limitations* : Any thoughts on the postprocessing of landmark points? For example, various methods postprocess the obtained coordinates to clean up the results.

typo: borader -> broader

**Justification Of Final Rating:**

I am happy to see benchmarking on additional datasets adding to the evidence of the robustness and out-of-the-box utility of the framework, giving state-of-the-art results among limited method comparison. This already is a wonderful contribution to the community, in line with the widely appreciated nnUnet framework.

I have improved the final rating from "weak accept" to "strong accept", but almost tempted to not do so. I would have appreciated seeing comparison with state-of-the-art in the given dataset, for transparency (also stated earlier in the preliminary rating justification). It is understandable that these sota methods go extra mile in their pipeline with say, pretraining and custom losses, but it would be interesting to see that the nnLandmark baseline out-of-the-box is as competitive and very close or even better in performance(may be in the discussion section and then in the appendix??). For example, based on reported numbers in [1], nnLandmark seems to do better on some landmarks (and close on others) compared to more sophisticated APGA[1] architecture in the LFC Dataset.

[1] Gong, Haifan, et al. "Fetal Cerebellum Landmark Detection Based on 3D MRI: Method and Benchmark." IEEE Journal of Biomedical and Health Informatics (2025).

**Justification Of The Preliminary Rating:**

The contributions stated by the paper are backed by the experiment and well motivated. The problem of reproducibility and out-of-the-box usability is demonstrated well. On the other hand, additional evidence of the robustness of the proposed baseline against other formulation of landmark prediction tasks such as direct coordinate prediction and state-of-the-art on specific datasets could have been provided.

**Questions To Address In The Rebuttal:**

Reformulation of landmark regression as a (gaussian blob) dense prediction task and consequent reuse of the widely validated nnUnet framework is very interesting. Could the authors discuss some of the associated limitations and discuss the possibility of tailoring the self-configuring framework for the landmark regression specific tasks such as landmark postprocessing, incomplete landmarks etc. ?

---

> ### Author Response · Authors · 2026-01-24
> **Postprocessing and Limitations**
>
> Many thanks to all reviewers for the constructive feedback and interesting questions.
>
> - nnUNet adaption. nnLandmark builds on the existing infrastructure of nnUNet and is compatible with all respective configurations, including 3d fullres (which is the default recommendation and was used in the current results), 3d lower and cascade. From experience and very preliminary experimentation, the cascade configuration didn’t provide any advantage compared to the 3d fullres configuration.
>
> - Specific landmark postprocessing. We focused on implementing a general out-of-the-box solution which delivers strong performance across a wide range of datasets without any specific postprocessing. At the same time, we see nnLandmark as a powerful starting point for further optimizations to achieve the best possible performance on a specific dataset. Similarly to nnU-Net, its open open design allows adjustments throughout the entire processing pipeline, including integrating specific postprocessing in the inference. In the light of our results, so far we didn't see the need for any postprocessing cleaning up the predictions.
>
> - Limitations. One limitation of our formulation of relying on dense predictions and the initial representation of the landmark labels as multi-label segmentation maps is that the landmarks must have a distance of at least 3 voxel, otherwise the segmentations would overlap. This issue could be solved by using multi-channel labels, however, at the cost of high RAM usage during data loading.
>
> - Incomplete landmarks. Currently, we derive the landmark coordinates by taking the channel-wise argmax, which assumes that all landmarks are present in each image, a standard assumption for anatomical landmark localization. However, this formulation can be straightforwardly adapted to instead find local maxima in the heatmaps. This would allow it to handle missing landmarks by only predicting the landmarks for which the local maximum reaches a certain confidence threshold. This threshold can be determined via the 5-fold cross-validation. It would further allow generally to predict a variable number of instances, making nnLandmark suitable for detecting small objects, which can be presented by coordinates. This is currently planned as an extension of nnLandmark.
>
> We further would like to note that we added two additional datasets to our evaluation, the FeTA biometry and PDDCA dataset, to additionally strengthen our contribution.

---

### Author Rebuttal · Authors · 2026-01-24

**Rebuttal:**

Many thanks to all reviewers.

CDFo
- We intentionally focus on a general solution without task-specific postprocessing; however, nnLandmark is an open framework and generally supports integrating post-hoc refinements.
- While our current decoding assumes all landmarks are present, it can be straightforwardly extended from channel-wise argmax to local-maximum detection with confidence thresholds, enabling missing landmarks and detecting a variable number of object instances in general; we plan to add this extension.

uEFD
- We added ablation studies on the robustness of our hyperparameter choice (Appendix B) and comprehensive qualitative examples (Appendix F).
- Regarding novelty, we show that a properly configured 3D U-Net can achieve state-of-the-art results, a novel and impactful finding for the landmark community, challenging a prevalent misconception that 3D U-Nets perform rather subpar.
nnLandmark is the first self-configuring framework for 3D landmark detection, which works reliably on a wide range of datasets and finally enables access to automated landmark detection with no need for expert knowledge.
Previously progress was hampered through subpar baselines and a lack of standardized frameworks to evaluate new ideas. nnLandmark closes this gap, while delivering state-of-the-art accuracy. In parallel to how nnU-Net has revolutionized 3D segmentation, nnLandmark is set to become the new default method and a catalyst for scientific innovation.

1sG2
- Our augmentation is implemented as dense resampling; representing landmarks as heatmaps allows applying exactly the same transformation as for the image, ensuring consistency under interpolation and non-linear deformations.
- The target spacing is selected automatically during preprocessing; it is set to the per-axis median spacing of the train data. This reflects the context–precision trade-off (lower resolution -> more context; higher resolution -> more precision); we adopt nnU-Net’s proven default, but users can override the target spacing as desired.
- Currently we don’t have a mechanism implemented for sub-voxel accuracy. If the landmarks were annotated in a higher resolution, a workaround can be to upsample the images. Further, heatmaps can allow sub-pixel precision by adding gaussian fitting and reporting the center location, which could be integrated into nnLandmark’s inference function.

To additionally strengthen our contribution we added two additional public datasets to the benchmark.

**Supporting Material:**

/attachment/d66f8e6f3df0b5f815a81434d3bde3347dc238b0.pdf

---

### Meta-Review · Area_Chair_amqQ · 2026-02-04

**Recommendation:** Accept (Poster)
**Confidence:** 5

**Metareview:**

There is a strong consensus amongst the reviewers to accept this work and I agree. In particular, the rebuttal appears to have clarified several points of confusion, making it a stronger submission.

---

### Decision · Program_Chairs · 2026-02-13

Accept (Poster)